# Growth, Safety and Tolerance in Infants Fed Rice Protein Hydrolysate Formula: The GRITO Randomised Controlled Trial

**DOI:** 10.3390/nu17010162

**Published:** 2024-12-31

**Authors:** Anaïs Lemoine, Antonio Nieto-García, María Nieto-Cid, Beatriz Espín-Jaime, Ángel Mazón, Hocine Salhi, Dimitrios Salamouras, Nicolas Kalach, Roser de Castellar-Sansó, Jesús Delgado Ojeda, Víctor Manuel Navas-López

**Affiliations:** 1Service de Nutrition et Gastro-Entérologie Pédiatrique, Hôpital Armand Trousseau AP HP, Sorbonne Université, 75571 Paris, France; anais.lemoine@aphp.fr; 2Unidad de Neumología y Alergia Infantil, Hospital Universitari y Politècnic La Fe, Instituto de Investigación Sanitaria La Fe, 46026 Valencia, Spain; antonio.nieto@me.com (A.N.-G.); mnietocid@gmail.com (M.N.-C.); amazon@comv.es (Á.M.); 3Sección de Gastroenterología, Hepatología y Nutrición Pediátrica, Hospital Infantil Virgen del Rocío, 41013 Sevilla, Spain; beatriz.espin.sspa@juntadeandalucia.es; 4Medical Affairs Department, Laboratoire Modilac, 92300 Levallois-Perret, France; hocine.salhi@sodilac.com; 5Service d’Allergologie et d’Anaphylaxie Pédiatrique, Hôpital Universitaire des Enfants Reine Fabiola, 1020 Bruxelles, Belgium; dimitri.salamouras@huderf.be; 6Groupement des Hôpitaux de l’Institut Catholique de Lille, Hôpital Saint Vincent de Paul, 59000 Lille, France; nicolas.kalach@gmail.com; 7Medical Affaires Department, Laboratorios Ordesa, Sant Boi de Llobregat, 08038 Barcelona, Spain; roser.decastellar@ordesalab.com (R.d.C.-S.); jesusojeda82@gmail.com (J.D.O.); 8Sección de Gastroenterología y Nutrición Infantil, Hospital Regional Universitario de Málaga, 29010 Málaga, Spain

**Keywords:** cow’s milk protein allergy, hydrolysed rice-based formula, tolerance, children, arsenic

## Abstract

**Background**: Hydrolysed rice formula (HRF) is tolerated by >90% of children with cow’s milk protein allergy (CMPA). However, concerns have been raised about potential suboptimal growth in infants fed HRF compared to those fed an extensively hydrolysed milk protein formula (eHF). **Aims**: To compare growth, safety and tolerance acquisition in infants with CMPA when fed HRF versus eHF. **Methods**: A multicentre prospective, randomised, double-blind, placebo-controlled food challenge trial was conducted with infants with CMPA. The infants received either HRF or eHF over a 12-month follow-up period. The primary outcome measure was the change from baseline over the study period in weight-for-length expressed as a Z-score. The secondary outcomes were other anthropometric measurements, tolerability and adverse events (AEs). **Results**: In total, 105 children were enrolled. The weight-for-length measurements were −0.01 (HRF) and −0.29 (eHF) at baseline and 0.29 and 0.05, respectively, at the last visit, with no significant between-group difference *(p* = 0.28; mixed-effects model). The Z-scores for other anthropometric variables indicated normal growth, with no significant between-group differences. In total, 29 potentially product-related AEs were reported (12 in the HRF group and 17 in the eHF group). A trend was observed toward a faster acquisition of tolerance in the HRF group (median age: 20.4 months) compared to the eHF group (16.3 months), but this was not statistically significant (*p* = 0.18). **Conclusions**: HRF demonstrated appropriate growth, acquisition of tolerance and a good safety profile in infants with CMPA, with no significant differences versus eHF. HRF could be considered as an appropriate option in the management of CMPA.

## 1. Introduction

Managing cow’s milk protein allergy (CMPA) involves the strict avoidance of cow’s milk protein and related products while ensuring adequate nutrition, typically achieved through specialised infant formulas. These include extensively hydrolysed cow’s milk protein formulas (eHF), amino acid formulas, hydrolysed rice formulas (HRF) or soy infant formulas. According to current practice guidelines from the European Society for Paediatric Gastroenterology, Hepatology and Nutrition (ESPGHAN), published in 2023 [1], cow’s milk-derived eHF is still the first choice for CMPA management, but HRF is now considered as a first-line alternative as well. Soy formulas may also be considered as an alternative, especially for economic, cultural and palatability reasons. However, using soy formulas is not recommended in Europe, especially in infants below six months of age due to the risk of co-allergy and the potential presence of phytoestrogens [1,2,3,4,5,6]. Amino acid formulas are reserved for more severe cases or in infants with impaired nutritional status, anaphylaxis and eosinophilic esophagitis [1]. While some individuals may tolerate mare’s or donkey’s milk, according to current ESPGHAN guidelines, the use of milk from other mammals with intact proteins is not advised in the management of CMPA, due to the significant homology between cow’s, sheep’s, and goat’s milk, which may lead to cross-reactivity in allergic patients [1]. Moreover, these options are costly and are not nutritionally adapted to meet the needs of paediatric patients with CMPA [1,7,8].

The relevance of using HRF lies in its potential positive impact on patient coverage by expanding management options for patients with CMPA, providing alternatives that respect individual beliefs and preferences. Allowing families to choose treatment options in line with their religious, social or emotional convictions promotes more patient-centred care, empowering them and increasing their satisfaction with the therapeutic process. Additionally, demonstrating that any available alternatives achieve both symptom resolution and tolerance acquisition ensures a safe and effective approach aligned with clinical standards.

Data from large, well-conducted randomised clinical trials concerning the relative benefits of HRF and eHF in infants with CMPA is limited. Although initial concerns had been raised regarding the possible low protein quality of rice-based formulas potentially impacting growth [9], as well as regarding the arsenic content of rice that may be present in HRF [10], the majority of studies have shown that infants fed HRF exhibit normal growth, both for healthy infants [11,12] and in infants with CMPA [3,13]. In addition, these studies have demonstrated the good tolerability of HRF in infants with CMPA. The consensus now is that HRF is an appropriate alternative to eHF for feeding infants with CMPA [14], and this is reflected in the 2023 ESPGHAN guidelines [1]. Nonetheless, more randomised clinical trials comparing eHF and HRF with the primary objective of assessing growth and safety are needed to ensure that physicians feel confident to use HRF in daily routine practice.

Acquiring tolerance to cow’s milk proteins is also an important goal in CMPA management. However, very few studies have evaluated tolerance acquisition for HRF. In a non-randomised study in infants with CMPA fed with different formulas [15], the rate of children acquiring oral tolerance after 12 months did not differ significantly (*p* = 0.26) between the group receiving eHF (43.6%) and the group receiving HRF (32.6%). In contrast, in a randomised study, Terracciano et al. found that infants and children with CMPA receiving HRF achieved tolerance earlier than their peers fed eHFs [16]. However, patient numbers in this study were very low (18–29 patients per group), and the infants had already been exposed to cow’s milk free formulas for variable amounts of time at inclusion. Further randomised clinical trials are clearly warranted to characterise tolerance acquisition in patients fed HRF compared to the other reference formulas.

The present study was initiated to compare outcomes between infants with CMPA fed with HRF and those fed with an eHF. The primary objective was to compare growth at 6, 9 and 12 months after inclusion into the study between the two randomised groups. Secondary objectives were to evaluate anthropometrics, protein status, safety, the acquisition of tolerance to cow’s milk proteins and the risk of arsenic exposure.

## 2. Materials and Methods

### 2.1. Study Design

This multicentre prospective, randomised, double-blind, placebo-controlled food challenge (DBPCFC) trial was performed from 2014 to 2019 across six hospital paediatric clinics in Spain, France and Belgium. The DBPCFC design is the “gold standard” for diagnosing a food allergy. Infants under 10 months old with a confirmed diagnosis of CMPA were randomly assigned to receive HRF or eHF for 12 months. Data were collected using an electronic case report form completed by each investigator. The design of the study is illustrated in Appendix A.

### 2.2. Study Population

Infants younger than 10 months old with confirmed CMPA diagnosed within two months prior to baseline (up to Grade II anaphylaxis [17]) were eligible. Diagnosis required a positive challenge on the DBPCFC [18] with cow’s milk, a positive specific IgE for cow’s milk protein (alpha-lactalbumin, beta-lactoglobulin, casein or whole milk) or a positive milk atopy patch test [19].

Additional inclusion criteria were singleton birth, gestational age from 37 to 42 weeks, birth weight ≥ 2.500 g and Apgar score > 7 [range 0–10] at 5 min post-partum. Written informed consent from parents or guardians was mandatory.

Exclusion criteria were infants with previous signs of allergy to any eHF, with a confirmed history of acute, severe, potentially life-threatening allergic reactions after the isolated accidental ingestion of cow’s milk (Grade III or higher), with a daily formula intake less than 100 mL, the presence of major congenital malformations or neonatal diseases, severe concurrent or chronic diseases, intrauterine growth retardation or neonatal infections. Breastfeeding was not an exclusion criterion.

Infants involved in other trials, those with missing parental written informed consent, those unable to adhere to the protocol and those exhibiting clinically relevant liver, kidney or hematological abnormalities were also excluded.

### 2.3. Investigational Products

The HRF (this formula was commercialised under two brand names: Blemil Plus^®^ hydrolysed rice, Laboratorios Ordesa, Barcelona, Spain and Modilac Expert Riz^®^, Laboratoires Modilac, Levallois-Perret, France) was developed for the dietary management of CMPA. It contains 100% partially hydrolysed rice protein supplemented with lysine and tryptophan in compliance with Directive 2006/141/EC. The eHF (Blemil Plus FH^®^; Laboratorios Ordesa) consisted of an extensively hydrolysed casein derived from cow’s milk. The composition of the formulas complied with requirements defined in European legislation (DE 2006/141/EC) at the time of the study.

### 2.4. Randomisation Procedures and Blinding

Subjects were enrolled consecutively with an identification number and randomly assigned to one of the two arms at a ratio of 1:1, with a block size of eight. The investigator was required to maintain a screening record provided by the clinical research organisation responsible for the operational management of the study, in which all screened subjects were registered. The date of enrolment, randomisation code and identification number of the subject were documented.

The masking of investigational products and randomisation assignment were performed centrally by the study sponsor. To ensure blinding, investigational products were provided in identical packaging, with only the batch number and expiry date. Investigational products to be used in subjects below the age of 6 months or in subjects between the ages of 6 to 12 months were packaged separately and provided to the study site pre-labelled with a code corresponding to the formula.

Investigators, support staff and the infants’ families were blinded to the product’s identity, known only to the clinical research organisation or study sponsor representative. Sealed envelopes (code breaks) containing randomisation assignments were provided to the Principal Investigator at each site. The code could only be broken in case of serious adverse events requiring the principal investigator to initiate an appropriate treatment. Investigators and study staff remained blinded to study treatment assignments until the statistical analysis was complete.

### 2.5. Study Procedures

At enrolment, parents were given diaries to record formula intake, tolerance, stool frequency/consistency and adverse events. At Visit 0, infants were randomly assigned to one of the two treatment arms. Parents were given diaries to record formula intake, tolerance, stool frequency/consistency and adverse events. Initially, the infants underwent a three-week elimination diet and did not start the allocated formula until a positive DBPCFC was confirmed at Visit 1. If the test was negative, the infant was withdrawn from the study. The remaining infants started their allocated formula at Visit 1. In cases of formula intolerance, infants were withdrawn from the study. Data were collected during follow-up visits 2 to 7 (at 1, 2, 3, 6, 9 and 12 months), including a full clinical assessment, anthropometric data, adverse events and concomitant medications (Appendix A). An open oral food challenge (OFC) was performed at Visits 5, 6 and 7. Infants with a negative challenge result were switched to standard formula and were followed until study completion. Urine and hair samples were taken at visits 0, 3 and 7 and blood samples at visits 0, 3, 5, 6 and 7. Urine and hair samples were obtained only from children exclusively or predominantly formula-fed.

### 2.6. Study Variables

Length (cm) and weight (kg) were measured at Visits 5, 6, and 7, and the length-for-weight ratio was calculated. The primary outcome measure was the change from baseline over the study period in weight-for-length expressed as a Z-score.

Secondary anthropometric outcome measures included change from baseline in length and weight, change from baseline in head circumference (cm), triceps skin fold thickness (mm), mid-arm circumference (cm), arm muscle area (cm^2^) and body mass index (BMI; kg/m^2^). Anthropometric measures were determined as absolute numbers or Z-scores. Reference values were obtained from the WHO MGRS study score [20], except for head circumference, for which Z-scores were calculated using 1990 British growth reference values [21].

Blood samples were drawn from all infants to determine the specific IgE for cow’s milk protein (Specific IgE CAP test Thermofisher^®^, Waltham, Massachusetts, United States), IgG, IgA, IgM, haemoglobin and haematocrit, alanine aminotransferase, urea, creatinine, albumin, ferritin and plasma amino acids. Urine samples were taken to measure creatinine and total and inorganic arsenic. Hair samples were taken to measure total arsenic, which was determined by the Institute of Agrochemistry and Food Technology (IATA) in Valencia, Spain.

Acquisition of tolerance was defined by a negative OFC and subsequent tolerance to standard cow’s milk, and the infant’s age at acquisition of tolerance was determined. Time to acquisition of tolerance was defined as the time between visit 1 and the switch to a cow’s milk formula. The cumulative number of infants who acquired tolerance was documented in three time periods during the follow-up period, at 6, 9 and 12 months (i.e., when the OFC was performed).

Digestive symptoms (including number of stools, regurgitations or episodes of colic) were identified from the parent diaries and were calculated for each between-visit interval.

All adverse events (AE) were documented by the investigators and listed by system-organ class and preferred term according to the MedDRA glossary (Version 26.1). These AEs were classified as serious or non-serious and potentially related or unrelated to the study formulas.

### 2.7. Statistical Analyses

Data were analysed on an intent-to-treat (ITT) basis. The ITT population included all subjects, regardless of whether they satisfied the entry criteria, the treatment received and subsequent withdrawal or deviation from the protocol. Missing data at individual time points were not replaced. Data at baseline were summarised using mean values with a standard deviation for normally distributed variables, median values with a range for continuous variables for non-normally distributed variables and frequency counts and percentages for categorical variables.

To achieve adequate statistical power, we calculated that 53 subjects would be needed in each group. This assumed an alpha risk of 0.05 and a power of 0.8 in a two-sided test to detect a statistically significant difference between groups of at least one unit in the primary outcome variable (change in Z score).

The primary outcome variables (change from baseline in Z-scores) were analysed using a mixed-effects model to quantify changes in weight and length trajectories at 12 months of follow-up adjusting for covariates. The covariates included were the type of formula, visit number and allergy type (IgE-mediated versus non-IgE-mediated). Specifically, random intercept and random slope models were implemented to account for individual variability in baseline Z-scores and the rate of change over time. Additionally, quadratic-order polynomial models were applied where the endpoint exhibited a non-linear pattern, allowing for a more accurate representation of the data trajectory. The models included both fixed effects (study arm and timepoint) and random effects (subject-level intercepts and slopes), enabling the evaluation of interaction effects between visit and group. This approach allowed for robust comparisons across study arms, adjusting for intra-individual variability and capturing statistically significant differences with precision.

Secondary anthropometric endpoints (triceps fold thickness, mid arm circumference and arm muscle area) were analysed using the same mixed-effects model as the primary outcome variable. For the longitudinal models for secondary endpoints (triceps fold thickness, mid arm circumference and arm muscle area), the time variable was corrected using “centred age” (age minus baseline mean age) because child growth is expected to be age-dependent. An analysis of anthropometric parameters was conducted as a worst-case scenario, excluding individuals with a negative OFC test at 9 and 12 months.

Tolerance acquisition was evaluated by Kaplan–Meier survival analysis and compared between groups using a Cox proportional hazard model.

For episodes of colic and regurgitations, an ordinal logistic regression model was fitted with the number of colics and regurgitations as the dependent variable and the visit and group as explanatory variables. Two-tailed *t*-tests and 95% confidence intervals were used for comparing number of stools at 6, 9 and 12 months. One-tailed *t*-tests and 95% confidence intervals were used for comparing arsenic levels in the population. Patients lost to follow-up were considered as still allergic at the end of the study (worst-case scenario). Statistical analyses were performed using Stata Statistical Software (Version 17) [22].

## 3. Results

Overall, 117 infants were enrolled and randomly assigned one of the study formulas. After randomisation, the parents of six infants refused to participate. During the run-in period between baseline (V1) and visit V2, DBPCFC was performed in all patients. Five infants with a negative DBPCFC and one with a severe reaction to the study formula during DBPCFC were withdrawn from the trial. The remaining 105 infants were included in the intention-to-treat analysis (50 assigned to the HRF group and 55 assigned to the eHF group) at baseline visit. A total of 89 infants completed the 6-month visit, 79 completed the 9-month visit and 71 completed the final study visit at 12 months (Appendix A). The baseline characteristics of the study population are presented in Table 1.

### 3.1. Primary Outcome Measure

The change in Z-scores across visits for the primary outcome of weight-for-length, as well as for weight and length independently, are presented by the study group in Figure 1. The Z-score for weight-for-length ratio increased significantly over the course of the study in patients from both treatment arms (*p* = 0.001 for the visit effect between baseline and 12 months in both groups). No significant difference in Z-score was observed between the two study groups (*p* = 0.28), and no “visit × group” interaction was observed (*p* = 0.15). Similar patterns were observed for both components of the primary outcome measure: for the weight Z-score, the visit effect was significant (*p* < 0.001), but not the group effect (*p* = 0.69) or the “visit × group” interaction (*p* = 0.86). For the length Z-score, the visit effect was significant (*p* = 0.04), but not the group effect (*p* = 0.64) or the “visit × group” interaction (*p* = 0.14). Minimal size effects were found for all the primary endpoints and visits (at 12 months, Cohen’s d = 0.015 for weight Z-score, d = 0.24 for length Z-score and d = 0.13 for weight-for-length Z score). In the mixed-effects model, no significant effect of type of formula and IgE status was observed, whereas the visit effect was significant. Full data on all anthropometric parameters and the sample size used for the calculation of the Z-score are presented by the study visit in Appendix A.

### 3.2. Secondary Anthropometric Outcome Measures

The change in Z-scores for BMI and head circumference across visits is presented by the study group in Figure 2. For BMI Z-score, the visit effect was significant (*p* < 0.0001), but not the group effect (*p* = 0.22) or the “visit × group” interaction (*p* = 0.11). For head circumference, no significant visit effect (*p* = 0.17), group effect (*p* = 0.50) or “visit × group” interaction (*p* = 0.90) were observed.

Minimal size effects were observed for all secondary endpoints and visits (at 12 months, Cohen’s d = 0.48 for BMI). In addition, the effect group and “age × group” interaction was not significant for mid-arm circumference (*p* = 0.21, *p* = 0.09), arm muscle area (*p* = 0.89, *p* = 0.99) or triceps skin fold (*p* = 0.45, *p* = 0.09, respectively) during the follow-up, whereas a significant effect of age was observed for mid-arm circumference (*p* < 0.001) and arm muscle area (*p* < 0.001) but not for triceps skin fold (*p* = 0.31) (Figure 3).

### 3.3. Tolerance to Cow’s Milk Proteins

During the follow-up period, tolerance to cow’s milk protein was documented in the 77 infants with reported age values (36 in the HRF group and 41 in the eHF group). At the end of the 12-month follow-up, 72.2% of infants in the HRF group and 53.7% of those in the eHF group acquired cow’s milk protein tolerance, with no significant difference between groups (*p* = 0.09; χ^2^ test). Kaplan–Meier survival curves for tolerance acquisition are presented in Figure 4. The median age at tolerance acquisition was 16.3 months in the HRF group and 20.4 months in the eHF group (*p* = 0.18; logrank test).

A subgroup analysis was performed in patients with IgE-positive and IgE-negative CMPA (Figure 5). The proportion of infants with IgE-positive CMPA who acquired tolerance during the study follow-up was 53.3% (16/30) in the HRF group and 37.9% (11/29) in the eHF group. Also, the proportion of infants with IgE-negative CMPA who acquired tolerance during the study follow-up was 36.0% (9/25) in the HRF group and 42.3% (11/26) in the eHF group. No significant difference in this proportion was observed between IgE-positive and IgE-negative children in either the HRF group (*p* = 0.20) or the eHF group (*p* = 0.74). In patients with IgE-positive CMPA, the median age at tolerance acquisition was 22.3 months in the eHF group and 16.9 months in the HRF group, with no significant difference between groups. In patients with IgE-negative CMPA, the median age at tolerance acquisition was 14.2 months in the eHF group and 14.6 months in the HRF group, with no significant difference between groups and a hazard ratio of 1.47 (95% CI: 0.46,4.72; *p* = 0.52).

### 3.4. Digestive Tolerance

Full data are presented in Appendix A. No differences were seen between study groups at any time point (group effect: *p* = 0.97 for regurgitations and *p* = 0.506 for colic). The incidence of regurgitations and of colic decreased over the follow-up period (time effect: *p* < 0.001 for regurgitations and *p* = 0.45 for colic).

### 3.5. Arsenic

Arsenic concentrations were low for both formulas. In hair, total arsenic concentrations were <0.07 mg/kg in all samples from both groups. The median arsenic concentration in urine was similar between groups, 2.24 [95%CI: 1.79–3.13] µg/L in the HRF group vs. 2.3 [95%CI: 1.56–2.63] µg/L in the eHF group (*p* = 0.48; Wilcoxon test).

### 3.6. Adverse Events

The frequency of reported adverse events (AEs) was comparable between the two groups (47 AEs in 26 patients in the HRF group and 51 AEs in 26 patients in the eHF group). One infant in the eHF group exhibited clinical intolerance to the product, as evidenced by an immediate allergic reaction, whereas no infant showed an allergic reaction to HRF.

Sixteen serious AEs were reported in both groups, occurring in three infants in the HRF group and four infants in the eHF group. None of these serious AEs were deemed related to the study treatment by the investigators. Fifteen of these events were classified as serious since they required hospitalisation, but all these serious AEs were resolved satisfactorily (Appendix A).

Twenty-nine adverse events considered potentially related to the product were reported in 15 patients (Appendix A). None of these events were considered as severe. Further details of the 29 potentially related AEs are presented in Appendix A.

## 4. Discussion

This study showed that infants in both the HRF and eHF groups exhibited normal growth over the 12-month follow-up period, with no significant differences in anthropometric measures. Recent guidelines consider HRF equivalent to eHF for feeding infants with CMPA [1,23]. Moreover, the results of this randomised controlled study confirmed the ability of an HRF to sustain normal growth, as shown in previous studies in healthy infants [11,12]. The nutritional quality of rice proteins is suitable for use in infant formulas since it is supplemented in certain amino acids that may be lacking in rice, typically lysine, threonine and tryptophan [14]. Taken together, all these outcomes suggest that physicians can prescribe, and parents can use, HRF to feed infants with CMPA with confidence and in expectation of an acquisition of tolerance to cow’s milk proteins.

Our findings challenge previous suggestions that rice formulas might lead to suboptimal growth compared to eHF or breast milk [9,24]. The American study [24] used a “rice juice” rather than a specific infant formula, and, for this reason, this study is difficult to interpret. The Italian study [9] was neither randomised nor blinded and enrolled a relatively small number of patients (15 fed HRF and 26 fed eHF). It reported lower infant weights during the 9–12 month and 12–18 month periods after treatment initiation, but at the end of the study at 24 months, weight gain was not significantly different between infants receiving eHF and those receiving HRF [9]. On the other hand, other larger studies have reported that infants fed HRF showed similar growth parameters to eHF [3,13]. These include two randomised, open-label trials, one of which enrolled 108 infants [3] and the other 92 infants [13], randomised to receive a characterised HRF, a standard eHF or a soy formula for a period of two years. Neither of these studies reported any difference in weight gain between the HRF and eHF formulas. In terms of numbers of infants treated, and the robustness of the study design, the balance of evidence is thus in favour of the normal growth of infants fed HRF.

Acquisition of tolerance is an important milestone in the outcome of children with CMPA, and is observed between the ages of 3 to 4 years in 80% of children in the majority of previous studies [25]. In the current study, tolerance was acquired before two years of age for more than half the infants, which suggests that avoiding cow’s milk protein entirely does not delay early tolerance acquisition. These results can be compared with previous observations from a previous study [16], which found that infants and children who had received HRF or a soy formulation for the dietary management of CMPA achieved tolerance significantly earlier than their peers receiving eHF. Nonetheless, it is important to bear in mind that not all eHF or HRF formulas are similar in their composition. For this reason, the findings of the present study only relate to the study products evaluated and the results cannot be generalised to all infant formulas currently available on the market.

Safety analysis showed comparable rates of adverse events in both groups, predominantly consisting of expected infant issues unrelated to treatment, such as fever and cough. These safety findings mirror those of previous research [13]. Arsenic exposure was low and comparable in both groups and was below recommended safety thresholds for maximum exposure [10,26,27]. Arsenic is a ubiquitous metalloid present at low concentrations in rocks, soil and natural ground water, mainly in inorganic forms. The assessment of the inorganic arsenic content of both HRF and eHF showed that the levels were below the maximum threshold of 0.02 mg/kg in the unreconsituted dry powder required by the European Union [26]. These results are also comparable with those published by Reche et al. [13].

The strengths of the study include its double-blind, randomised controlled design and confirmation of CMPA with a double-blind oral food challenge. Moreover, the primary outcome measure was an objective one, determined in the same study centre using the same protocol at each visit. The limitations of this study include the relatively small sample size and the enrolment of infants at a relatively older age, potentially influenced by prior infant formula consumption. A larger sample size would have allowed for a more detailed analysis, including the assessment of covariates that may influence the development of tolerance, such as breastfeeding and the better control of potential centre effects. Additionally, the study design did not permit a formal demonstration of the non-inferiority HRF compared to eHF. Although there was a small imbalance between study groups (50 vs. 55), the randomisation process and the comparable baseline characteristics of the groups minimise the likelihood of significant bias. Further research on tolerance acquisition in infants with CMPA fed with HRF is still warranted. In the present study, we did not identify any relevant difference in time to tolerance acquisition in the full study population, although a trend to more rapid tolerance acquisition was observed in the subgroup of patients who were IgE-positive. Interestingly, in the analysis of CMPA and tolerance acquisition performed in the European EuroPrevall study [28], IgE-negative patients more frequently acquired tolerance than IgE-positive patients by the age of 2 years, although data on the use of cow’s milk-free formulas were not collected. Neither the MICMAC II study [16] nor the present subgroup analysis evaluated sufficient patients to resolve this question, and a larger dedicated study would be important to perform in order to resolve the issue and to confirm any possible associations between the choice of formula, IgE status and time to tolerance acquisition. It would also be of interest to evaluate whether prior breastfeeding has any effect on tolerance.

## 5. Conclusions

This study confirms that HRF can be used with confidence as a first-line alternative to cow’s milk for feeding infants with CMPA, resulting in normal growth, the acquisition of tolerance and no clinically relevant safety issues. This will expand management options for patients and physicians and allow parents a choice in the nutrition of their children with CMPA.

Future research should focus on larger, multicentre studies with a broader age range at enrolment, detailed data on breastfeeding and designs aimed at formally evaluating the non-inferiority of HRF in diverse CMPA populations.

## Figures and Tables

**Figure 1 nutrients-17-00162-f001:**
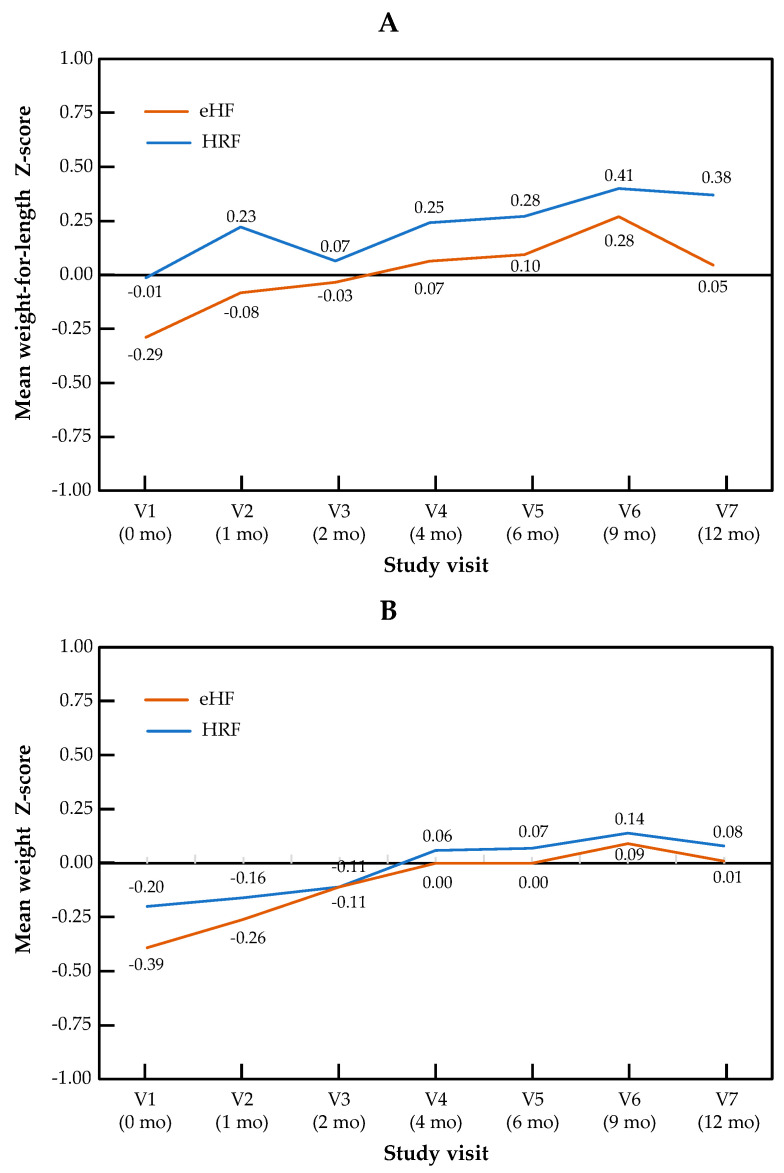
Growth trajectories for weight-for-length (**A**), weight (**B**) and length (**C**) in the HRF and eHF groups over the study period (intention-to-treat population).

**Figure 2 nutrients-17-00162-f002:**
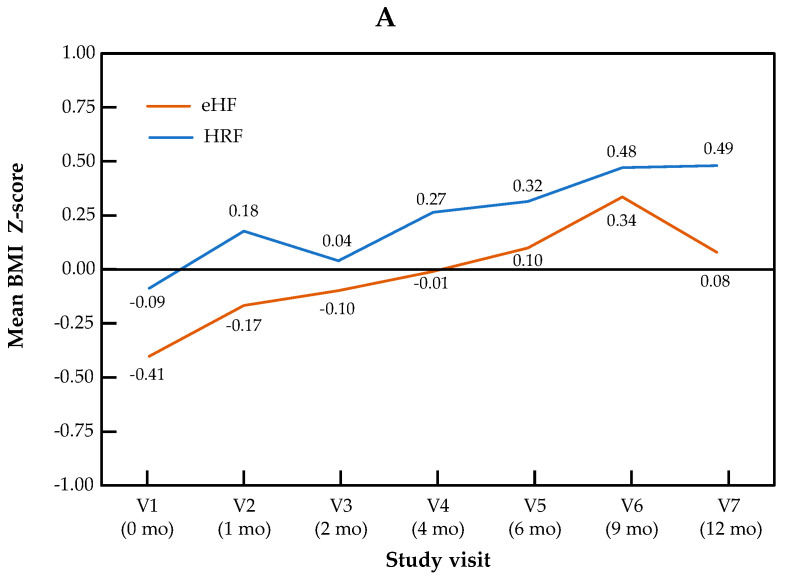
Growth trajectories for BMI (**A**) and head circumference (**B**) Z-scores in the HRF and eHF groups over the study period (intention-to-treat population).

**Figure 3 nutrients-17-00162-f003:**
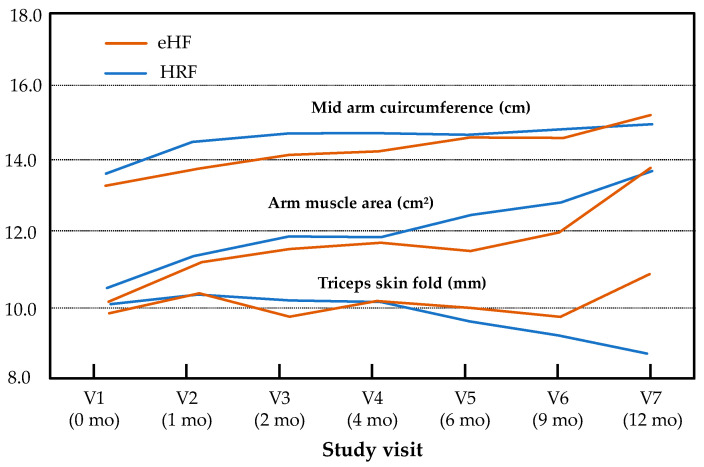
Mid-arm circumference, arm muscle area and triceps skin fold over the study period (intention-to-treat population). Results are presented as estimated means for normally distributed variables (mid-arm circumference, triceps skin fold) and medians for non-normally distributed variables (arm muscle area).

**Figure 4 nutrients-17-00162-f004:**
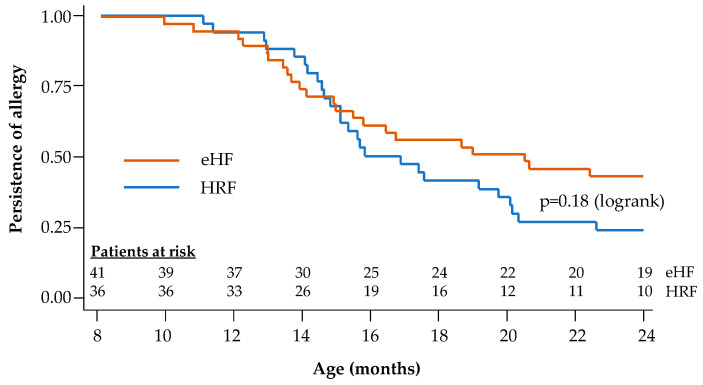
Kaplan–Meier survival curves for acquisition of cow’s milk protein tolerance: Cox regression model.

**Figure 5 nutrients-17-00162-f005:**
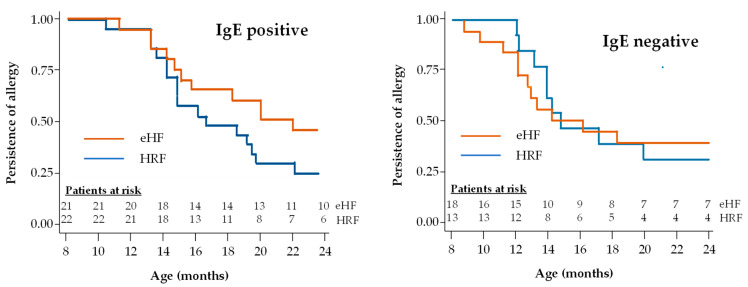
Kaplan–Meier survival curves for acquisition of cow’s milk protein tolerance in infants with IgE-positive and IgE-negative CMPA.

**Table 1 nutrients-17-00162-t001:** Baseline characteristics of the study population.

Variable	HRF	eHF
Age at baseline (months) Mean ± SD	*n* = 50 6.8 ± 2.2	*n* = 55 6.4 ± 2.4
Gender, *n* (%) Male	*n* = 50 24 (48.0%)	*n* = 55 35 (63.6%)
Weight (kg) Mean ± SD	*n* = 44 7.686 ±1.274	*n* = 48 7.406 ± 1.314
Length (cm) Mean ± SD	*n* = 44 67.2 ± 4.5	*n* = 48 66.9 ± 5.6
BMI (kg/m^2^) Mean ± SD	*n* = 44 16.9 ± 1.5	*n* = 48 16.4 ± 1.2
Head circumference (cm) Mean ± SD	*n* = 44 43.4 ± 2.4	*n* = 44 43.5 ± 3.3
Triceps skin fold (mm) Mean ± SD	*n* = 44 10.11 ± 2.55	*n* = 44 9.92 ± 2.66
Mid arm circumference (cm) Mean ± SD	*n* = 44 13.8 ± 1.9	*n* = 44 13.3 ± 1.8
Arm muscle area (cm^2^) Mean ± SD	*n* = 44 10.6 ± 3.6	*n* = 44 10.3 ± 3.7
CMP IgE positive (*n*, %)	*n* = 50 27 (54.0%)	*n* = 55 29 (52.7%)
Breastfed infants (*n*, %)	*n* = 50 27 (54.0%)	*N* = 55 25 (45.5%)

CMP: cow’s milk protein; eHF: extensively hydrolysed formula; HRF: hydrolysed rice formula; IgE: immunoglobulin E; SD: standard deviation.

## Data Availability

The data that support the findings of this study are not openly available due to reasons of sensitivity. However, they are available from the corresponding author upon reasonable request.

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
