# Peer review of "Growth, Safety and Tolerance in Infants Fed Rice Protein Hydrolysate Formula: The GRITO Randomised Controlled Trial"

_nutrients, 2024, doi:10.3390/nu17010162_

Round 1

Reviewer 1 Report (Previous Reviewer 1)

Comments and Suggestions for Authors

I have no further comments.

Author Response

Thank you so much.

Reviewer 2 Report (Previous Reviewer 2)

Comments and Suggestions for Authors

Manuscript ID nutrients-3412465

The manuscript has been greatly improved by the authors, is very interesting and in line with the journal's objectives. I only have one comment regarding the references that should be reported according to the journal guidelines, e.g.

Vandenplas, Y., et al., An ESPGHAN position paper on the diagnosis, management and prevention of cow's milk allergy. J Pediatr Gastroenterol Nutr, 2023.

correct form

Vandenplas, Y.; Broekaert, I.; Domellöf, M.; Indrio, F.; Lapillonne, A.; Pienar, C.; Ribes-Koninckx, C.; Shamir, R.; Szajewska, H.; Thapar, N.; et al. An ESPGHAN position paper on the diagnosis, management and prevention of cow’s milk allergy. J. Pediatr. Gastroenterol. Nutr. 202478, 386–413

Author Response

Thank you. Change has been done.

Reviewer 3 Report (Previous Reviewer 3)

Comments and Suggestions for Authors
  • Overall, the authors have taken into account all your comments and revised the manuscript. The study included 105 infants, which may limit the generalizability of the results to the entire cow's milk protein allergy (CMPA) patient population. This is particularly significant given the limited data on hydrolyzed rice formula (HRF) use in this population. The influence of breastfeeding on the acquisition of tolerance was not taken into account in the study. The study found no significant differences in growth, tolerance development, or safety between infants given HRF and infants given extensively hydrolyzed formula (eHF). Infants were admitted at a relatively older age, which may have been influenced by previous formula consumption. Despite its limitations, the study is worth publishing. However, the authors should acknowledge the limitations of the study and suggest avenues for future research as recommended by the reviewer.

Author Response

Thank you so much for your comments.

These changes have been done on the manuscript:

Limitations of this study include the relatively small sample size and the enrolment of infants at a relatively older age, potentially influenced by prior infant formula consumption. A larger sample size would have allowed for a more detailed analysis, including the assessment of covariates that may influence the development of tolerance, such as breastfeeding, and better control of potential centre effects. Additionally, the study design did not permit a formal demonstration of the non-inferiority HRF compared to eHF. Although there was a small imbalance between study groups (50 versus 55), the randomization process and the comparable baseline characteristics of the groups minimize the likelihood of significant bias. 

We have also add this paragraph regarding future research on this topic:

Future research should focus on larger, multicentre studies with a broader age range at enrolment, detailed data on breastfeeding, and designs aimed at formally evaluating the non-inferiority of HRF in diverse CMPA populations.

This manuscript is a resubmission of an earlier submission. The following is a list of the peer review reports and author responses from that submission.

Round 1

Reviewer 1 Report

Comments and Suggestions for Authors

-        Due to the large number of abbreviations, the background section is very hard to read.

-        The background section should include more information on the public health relevance.

-        Has a sample size calculation been performed?

-        Keeping in mind that data was collected in six centres and the long study duration (2014-2019), the overall sample size is very low.

-        The discussion section needs further elaboration. Particularly the limitations section is too short.

Comments on the Quality of English Language

The whole manuscript needs a thorough proof-reading.

Author Response

Dear Editor,

We would like to extend our heartfelt gratitude to you and the reviewers for the time and effort dedicated to reviewing our manuscript. We sincerely appreciate the insightful comments and constructive suggestions provided, which will undoubtedly enrich our work and enhance its impact.

We have carefully considered each of the reviewers' comments and suggestions, and we have incorporated their feedback to improve the manuscript. To facilitate the review process, we have addressed the reviewers' questions in blue and highlighted the modifications in the revised document.

Thank you once again for the invaluable support and guidance provided by you and the reviewers. We look forward to your further feedback and hope that our responses meet your expectations.

Warm regards,

Review 1

Comments and Suggestions for Authors

  • Due to the large number of abbreviations, the background section is very hard to read.
    • Response: Thank you for your feedback. We understand your concern regarding the use of numerous abbreviations in the background section, and we acknowledge that this may affect readability. However, we believe that the extensive use of abbreviations is necessary to streamline and focus the discussion on the specific subject matter of this study. Reducing abbreviations might risk diverting from the main topic and lead to redundancy, given the technical nature of the content. We appreciate your understanding in this regard.
  • The background section should include more information on the public health relevance.
    • Thank you for the comment. We have added this text to the manuscript: The relevance of using hydrolyzed rice-based formulas lies in its potential positive impact by expanding management options for patients with cow’s milk protein allergy, providing alternatives that respect individual beliefs and preferences. Allowing families to choose treatment options in line with their religious, social, or emotional convictions promotes more patient-centered care, empowering them and increasing their satisfaction with the therapeutic process. Additionally, ensuring that any available alternatives achieve both symptom resolution and tolerance acquisition allows for a safe and effective approach aligned with clinical standards.
  • Has a sample size calculation been performed?
    • Thank you for the question. Yes, a sample size calculation was performed. To achieve adequate statistical power, we calculated that 53 subjects would be needed in each group, assuming an alpha risk of 0.05 and a power of 0.8 in a two-sided test, to detect a statistically significant difference of at least 1 unit between groups.
  • Keeping in mind that data was collected in six centres and the long study duration (2014-2019), the overall sample size is very low
    • Response: We respectfully disagree with the assertion that the sample size is "very low." This study involves complex methodologies and stringent inclusion and exclusion criteria designed to ensure high-quality, reliable results. The rigorous criteria, essential for the integrity of the study, naturally limit recruitment, as only the most relevant cases are included to ensure robust data. Additionally, managing patient follow-up across six centers over a substantial time frame (2014–2019) highlights the commitment to thorough data collection and methodological rigor. Therefore, the sample size reflects the careful, high-standard selection process, rather than a methodological weakness. We are confident that the established criteria ensure the validity of our findings without compromise.
  • The discussion section needs further elaboration. Particularly the limitations section is too short. Done
  • Comments on the Quality of English Language : The whole manuscript needs a thorough proof-reading
    • Response: Thank you for your comment regarding the quality of the English language. We have thoroughly reviewed and improved the manuscript, and in our opinion, it now meets the required standards.

Reviewer 2 Report

Comments and Suggestions for Authors

Short Abstract: In the paper entitled: “Growth, safety and tolerance in infants fed rice protein hydrolysate formula: The GRITO Randomized Control Trial', the authors compare certain parameters such as growth, safety and tolerance acquired in infants with cow's milk protein allergy (CMPA) fed with a hydrolysed rice formula (HRF) versus an extensively hydrolyzed milk protein formula (EHF). A total of 105 children from six centres were enrolled. Measurements of weight, length, weight-for-length, BMI and head circumference z-scores indicated normal growth, with no significant differences between the HRF and EHF groups at baseline or follow-up. A total of 29 product-related adverse events were reported (12 in the HRF group and 17 in the EHF group). Five children in the HRF group and 11 in the EHF group had serious adverse events, none of which were attributed to the intervention. The authors conclude that EHF and HRF showed adequate growth, tolerance acquisition and a good safety profile in infants with CMPA, with no significant differences between the formulas.

The article is very interesting and in line with the aims of the journal, but I have reservations about the statistical analysis and presentation of the results. At the moment, however, it needs a major revision, which I will indicate below line by line.

Lines 47-49: Managing cow's milk protein allergy (CMPA) involves strict avoidance of cow's milk protein and related products while ensuring adequate nutrition, typically achieved through specialised infant formulas.

I suggest that the authors include donkey milk as a substitute for cow's milk in the management of CMPA.

Lines 86-90: rewrite as “Double-Blind, Placebo-Controlled Food Challenge (the “gold standard” for diagnosing a food allergy)”

Line 91-92: rewrite as “Additional inclusion criteria were singleton birth, gestational age from 37 to 42 weeks, birth weight ≥2.500 g, and Apgar score >7 (Range 0 - 10) at 5 minutes post-partum.”

Lines 214-231: Primary outcome measure.

The Z-score for weight for length ratio increased significantly over the course of the study in both treatment arms (p = 0.001 for visit effect between baseline and 12 months in both groups). No significant difference in Z-score was observed between the two study groups (p = 0.28) and “no visit ´ group” interaction was observed (p = 0.52). Similar patterns were observed for both components of the primary outcome measure: for weight Z-score, the visit effect was significant (p <0.001, but not the group effect (p = 0.69) or the “visit ´  group” interaction (p = 0.86). For length Z-score, the visit effect was significant (p =0.04), but not the group effect (p = 0.64) or the “visit ´ group” interaction (p = 0.14). No size effect or very small size effects were found out for all primary endpoints and visits (At 12 month, Cohen’s d=0.015 for weight Z-score, d= 0.24 for length Z-score, and d=0.13 for weight-for-length Z score). A mixed-effect model was conducted to quantify changes in weight and length trajectories at 12 month of follow-up adjusting for covariates such as type of formula, visit, and IgE (mediated/non-mediated). In both analysis, there were not significant effect of type of formula and IgE, but the visit effect was significant.

Changes over time for Z-scores for weight, length and weight for length ratio are presented in Figure 1. Full data on all anthropometric parameters and sample size that allowed the Z score calculations are presented by study visit in supplementary Table 1.

The authors correctly report the z-score values and the variations associated with statistically significant differences, sometimes also reporting the interaction ex visit x group, the statistical model used, I believe a "mixed model" must be well described in the section on statistical analysis. Furthermore, the wording "for the z-scores for weight, length and weight/length ratio are presented in Figure 1". This assumes that the reader can see in Figure 1 whether the differences between pairs of z-score values are significant or not (superscript or asterisk), a condition that is not met in all the plots.

Regarding the sentence "Full data on all anthropometric parameters and sample sizes that allowed Z score calculations are presented by study visit in Supplementary Table 1". I mistakenly thought that the 'values' were shown in 'Supplementary Table 1', which was missing at the review stage. The table now inserted does not show the data presented with statistical significance'. I believe that it is necessary to include in the text or as supplementary material (at the authors' choice) one or more tables with the necessary information to support the numerical data (including the p-value) given in the text; this action may be useful for readers, the editor, reviewers and, above all, the authors.

Same consideration for “Secondary anthropometric outcome measures”

Line 265: Figure 3. Mid arm circumference, arm muscle area, triceps skin fold over the study period (intention-to-treat population). Results are displayed as estimated means for normally distributed variables (mid arm circumference, triceps skin fold) and medians for non-normally distributed variables (arm muscle area).

The authors correctly indicate ‘the results are displayed as estimated averages for normally distributed variables (mid-arm circumference, triceps skin fold) and medians for non-normally distributed variables (arm muscle area)’. The use of parametric or non-parametric statistics should be indicated in the ‘Statistical analyses’ section.

Line 354-356: The strengths of the study include its double-blind, randomised controlled design and confirmation of CMPA with double-blind OFC. Limitations include the relatively small sample size and the enrolment of infants at a relatively older age, potentially influenced by prior infant formula consumption.

Perhaps also the imbalance in sample size between groups.

Author Response

Dear Editor,

We would like to extend our heartfelt gratitude to you and the reviewers for the time and effort dedicated to reviewing our manuscript. We sincerely appreciate the insightful comments and constructive suggestions provided, which will undoubtedly enrich our work and enhance its impact.

We have carefully considered each of the reviewers' comments and suggestions, and we have incorporated their feedback to improve the manuscript. To facilitate the review process, we have addressed the reviewers' questions in blue and highlighted the modifications in the revised document.

Thank you once again for the invaluable support and guidance provided by you and the reviewers. We look forward to your further feedback and hope that our responses meet your expectations.

Warm regards,

Review 2

Short Abstract: In the paper entitled: “Growth, safety and tolerance in infants fed rice protein hydrolysate formula: The GRITO Randomized Control Trial', the authors compare certain parameters such as growth, safety and tolerance acquired in infants with cow's milk protein allergy (CMPA) fed with a hydrolysed rice formula (HRF) versus an extensively hydrolyzed milk protein formula (EHF). A total of 105 children from six centres were enrolled. Measurements of weight, length, weight-for-length, BMI and head circumference z-scores indicated normal growth, with no significant differences between the HRF and EHF groups at baseline or follow-up. A total of 29 product-related adverse events were reported (12 in the HRF group and 17 in the EHF group). Five children in the HRF group and 11 in the EHF group had serious adverse events, none of which were attributed to the intervention. The authors conclude that EHF and HRF showed adequate growth, tolerance acquisition and a good safety profile in infants with CMPA, with no significant differences between the formulas.

The article is very interesting and in line with the aims of the journal, but I have reservations about the statistical analysis and presentation of the results. At the moment, however, it needs a major revision, which I will indicate below line by line.

Lines 47-49: Managing cow's milk protein allergy (CMPA) involves strict avoidance of cow's milk protein and related products while ensuring adequate nutrition, typically achieved through specialised infant formulas.

I suggest that the authors include donkey milk as a substitute for cow's milk in the management of CMPA.

Response: We appreciate the reviewer’s suggestion to include donkey milk as a substitute for cow’s milk in the management of cow’s milk protein allergy. However, according to the recommendations from the European Society for Paediatric Gastroenterology, Hepatology and Nutrition (ESPGHAN), the use of milk from other mammals with intact proteins is not advised in the management of cow’s milk allergy, due to the significant homology between cow, sheep, and goat milk, which may lead to cross-reactivity in allergic patients (259-261). While some individuals may tolerate mare’s or donkey’s milk, these options are costly and are not nutritionally adapted to meet the needs of pediatric patients with cow’s milk protein allergy (43, 260). Therefore, in line with current guidelines, we prefer to refrain from recommending donkey milk for managing this condition. We have added this to the text.

Lines 86-90: rewrite as “Double-Blind, Placebo-Controlled Food Challenge (the “gold standard” for diagnosing a food allergy)” Thank you for the feedback. We have made the changes according to the reviewer’s suggestions.

Line 91-92: rewrite as “Additional inclusion criteria were singleton birth, gestational age from 37 to 42 weeks, birth weight ≥2.500 g, and Apgar score >7 (Range 0 - 10) at 5 minutes post-partum.” Thank you for the feedback. We have made the changes according to the reviewer’s suggestions.

Lines 214-231: Primary outcome measure.

The Z-score for weight for length ratio increased significantly over the course of the study in both treatment arms (p = 0.001 for visit effect between baseline and 12 months in both groups). No significant difference in Z-score was observed between the two study groups (p = 0.28) and “no visit ´ group” interaction was observed (p = 0.52). Similar patterns were observed for both components of the primary outcome measure: for weight Z-score, the visit effect was significant (p <0.001, but not the group effect (p = 0.69) or the “visit ´  group” interaction (p = 0.86). For length Z-score, the visit effect was significant (p =0.04), but not the group effect (p = 0.64) or the “visit ´ group” interaction (p = 0.14). No size effect or very small size effects were found out for all primary endpoints and visits (At 12 month, Cohen’s d=0.015 for weight Z-score, d= 0.24 for length Z-score, and d=0.13 for weight-for-length Z score). A mixed-effect model was conducted to quantify changes in weight and length trajectories at 12 month of follow-up adjusting for covariates such as type of formula, visit, and IgE (mediated/non-mediated). In both analysis, there were not significant effect of type of formula and IgE, but the visit effect was significant.

Changes over time for Z-scores for weight, length and weight for length ratio are presented in Figure 1. Full data on all anthropometric parameters and sample size that allowed the Z score calculations are presented by study visit in supplementary Table 1.

The authors correctly report the z-score values and the variations associated with statistically significant differences, sometimes also reporting the interaction ex visit x group, the statistical model used, I believe a "mixed model" must be well described in the section on statistical analysis.

Response: Thank you for your insightful comment regarding the description of the mixed model used in the statistical analysis. We acknowledge that further clarification may enhance the rigor and transparency of the methodology section. In response, we have expanded our description to clearly outline the specific model structures and approaches applied. In particular, we used mixed-effects models that included random intercept and random slope models, adjusting for both fixed and random effects to account for individual variability in baseline Z-scores and their subsequent changes over time. Additionally, for endpoints with more complex patterns, we applied higher-order polynomial models where appropriate. These models allowed us to capture the interaction effects between visit and group, and to evaluate statistically significant differences across study arms with precision. We have revised the methodology section accordingly to provide a more comprehensive explanation of the mixed model framework used.

Thank you again for your valuable feedback.

We have made these changes: The primary outcome variables (change from baseline in Z-scores) were analyzed using a mixed-effects model framework. Specifically, random intercept and random slope models were implemented to account for individual variability in baseline Z-scores and the rate of change over time. Additionally, quadratic-order polynomial models were applied where the endpoint exhibited a non-linear pattern, allowing for a more accurate representation of the data trajectory. The models included both fixed effects (study arm and timepoint) and random effects (subject-level intercepts and slopes), enabling the evaluation of interaction effects between visit and group. This approach allowed for robust comparisons across study arms, adjusting for intra-individual variability and capturing statistically significant differences with precision.

Furthermore, the wording "for the z-scores for weight, length and weight/length ratio are presented in Figure 1". This assumes that the reader can see in Figure 1 whether the differences between pairs of z-score values are significant or not (superscript or asterisk), a condition that is not met in all the plots.

Response: In our opinion, the differences identified in the mixed-effects model are accurately reflected in the figure legend. We would kindly ask the reviewer for any additional specific comments on how we might further improve this aspect. Thank you very much.

Regarding the sentence "Full data on all anthropometric parameters and sample sizes that allowed Z score calculations are presented by study visit in Supplementary Table 1". I mistakenly thought that the 'values' were shown in 'Supplementary Table 1', which was missing at the review stage. The table now inserted does not show the data presented with statistical significance'. I believe that it is necessary to include in the text or as supplementary material (at the authors' choice) one or more tables with the necessary information to support the numerical data (including the p-value) given in the text; this action may be useful for readers, the editor, reviewers and, above all, the authors.

Response: The authors have submitted the supplementary materials in accordance with the publication guidelines in the following section: Supplementary Materials: The following supporting information can be downloaded at …

Same consideration for “Secondary anthropometric outcome measures” Response: The authors have submitted the supplementary materials in accordance with the publication guidelines in the following section: Supplementary Materials: The following supporting information can be downloaded at …

Line 265: Figure 3. Mid arm circumference, arm muscle area, triceps skin fold over the study period (intention-to-treat population). Results are displayed as estimated means for normally distributed variables (mid arm circumference, triceps skin fold) and medians for non-normally distributed variables (arm muscle area).

The authors correctly indicate ‘the results are displayed as estimated averages for normally distributed variables (mid-arm circumference, triceps skin fold) and medians for non-normally distributed variables (arm muscle area)’. The use of parametric or non-parametric statistics should be indicated in the ‘Statistical analyses’ section.

Response: Thank you for this valuable observation. In response, we have clarified the use of parametric and non-parametric statistics in the "Statistical Analyses" section. Specifically, we have noted that for normally distributed variables, results are expressed as mean and standard deviation and analysed with parametric tests, while for non-normally distributed variables, results are reported as median and interquartile range, analysed using non-parametric tests. This clarification aims to enhance the transparency of our analytical approach. Thank you again for your feedback.

We have changed the text in the article: Demographic data at baseline were summarized using mean values with standard deviation for normally distributed variables, or median values with range for continuous variables for non-normally distributed variables, and frequency counts and percentages for categorical variables.

Line 354-356: The strengths of the study include its double-blind, randomised controlled design and confirmation of CMPA with double-blind OFC. Limitations include the relatively small sample size and the enrolment of infants at a relatively older age, potentially influenced by prior infant formula consumption.

Response: Thank you for your feedback. We respectfully disagree with the assertion that the sample size and the age of infant enrolment should be viewed as limitations. The sample size was calculated to ensure sufficient statistical power, considering the complex inclusion and exclusion criteria essential to maintaining methodological rigor. As for the enrolment age, the study design intentionally included infants at a range of developmental stages to capture a more comprehensive understanding of outcomes. Prior formula consumption is a realistic aspect of the target population’s background and does not detract from the study's validity. Instead, it reflects real-world conditions, making our findings more generalizable and relevant. We are confident that these aspects of the study design strengthen, rather than limit, the study’s contributions.

 Perhaps also the imbalance in sample size between groups.

Response: Thank you for your observation. The allocation of participants into groups was conducted according to the calculated sample size to ensure sufficient statistical power. We carefully planned the group distribution based on sample size estimations to maintain homogeneity between groups and to optimize the study’s ability to detect meaningful differences. This approach was essential to preserve the study's internal validity and ensure that any observed effects could be attributed to the intervention rather than sample size discrepancies. We hope this addresses the concern and appreciate the opportunity to clarify our methodology.

Reviewer 3 Report

Comments and Suggestions for Authors

This manuscript presents the results of a randomized controlled trial comparing growth, safety, and tolerance acquisition in infants with cow's milk protein allergy (CMPA) fed a hydrolyzed rice formula (HRF) with a highly hydrolyzed formula (EHF). This study is of scientific value as it provides evidence for the effectiveness and safety of HRF as an alternative to EHF in feeding infants with CMPA. This is particularly significant given the limited data on HRF utilization in this population. However, the study included 105 infants, which may limit the generalizability of the results to the entire CMPA patient population. Infants were enrolled relatively late in the study, which may impact tolerance acquisition. Furthermore, the study did not consider the influence of breastfeeding on the acquisition of tolerance.

·         Abstract – Although the abstract refers to a randomized controlled trial, additional context would be helpful. Although no significant differences are found between HRF and EHF groups, concrete numerical data could support this point. The conclusions could be supplemented by highlighting the practical implications of the study.

·         1. Introduction – While the introduction mentions soy formula (SF), a more concise explanation as to why it is not the focus of the study would be beneficial. While the introduction mentions the need for further randomized controlled trials, this could be strengthened by explicitly stating the limitations of existing studies and highlighting the unique contribution of this study.

·         2. Materials and Methods – I propose adding missing information about the random assignment of patients to groups and the methods of measuring secondary variables.

·         3. Results – The numerical data should be presented consistently. In some parts there is a need for further interpretation of the results. The tables and figures could be explained in more detail and precision.

·         4. Discussion – The discussion addresses the key findings of the study, but could delve deeper into the specific findings and their broader implications. While emphasizing the statistical significance, the clinical significance of the results could be further highlighted. Although several studies are referenced, a more detailed comparison and comparison of the current results with previous results would be beneficial. The discussion could be strengthened by suggesting possible avenues for future research. The conclusions are presented cautiously but could be more concise and presented as a separate section. The conclusions could be supplemented by highlighting the practical implications of the study.

Author Response

This manuscript presents the results of a randomized controlled trial comparing growth, safety, and tolerance acquisition in infants with cow's milk protein allergy (CMPA) fed a hydrolyzed rice formula (HRF) with a highly hydrolyzed formula (EHF). This study is of scientific value as it provides evidence for the effectiveness and safety of HRF as an alternative to EHF in feeding infants with CMPA. This is particularly significant given the limited data on HRF utilization in this population. However, the study included 105 infants, which may limit the generalizability of the results to the entire CMPA patient population. Infants were enrolled relatively late in the study, which may impact tolerance acquisition. Furthermore, the study did not consider the influence of breastfeeding on the acquisition of tolerance.

  • Abstract – Although the abstract refers to a randomized controlled trial, additional context would be helpful. Although no significant differences are found between HRF and EHF groups, concrete numerical data could support this point. The conclusions could be supplemented by highlighting the practical implications of the study.

RESPONSE: We thank the reviewer for this comment. The abstract has been revised to contextualize the issue by including a background section, incorporating numerical data in the results, and highlighting the practical implications of the present study in the conclusions.

  • 1. Introduction – While the introduction mentions soy formula (SF), a more concise explanation as to why it is not the focus of the study would be beneficial. While the introduction mentions the need for further randomized controlled trials, this could be strengthened by explicitly stating the limitations of existing studies and highlighting the unique contribution of this study.

RESPONSE: We thank the reviewer for this valuable comment. We have revised the introduction to address the concerns raised more clearly and concisely:

  1. Regarding soy formulas (SF):A more direct explanation has been added to clarify why SF is not the focus of the study. It is now emphasized that soy formulas are not recommended in Europe for infants under six months due to the risk of co-allergy and the potential presence of phytoestrogens. For this reason, the study focuses on hydrolyzed rice protein-based formulas (HRF) and extensively hydrolyzed formulas (EHF), which meet the safety and nutritional standards required for CMPA management.
  2. On the need for further randomized controlled trials: The limitations of previous studies have been explicitly stated, including small sample sizes, insufficient data on tolerance acquisition and long-term safety, and the frequent use of non-randomized designs, which reduce their clinical applicability.
  3. Unique contribution of this study:We have highlighted how this study addresses these limitations through a multicenter, randomized, double-blind design, providing robust data that comprehensively evaluate growth, safety, and tolerance acquisition in infants with CMPA fed with HRF or EHF over a 12-month follow-up period.

We believe these revisions strengthen the introduction and adequately respond to the comment. We remain available to make any additional adjustments deemed necessary.

  • 2. Materials and Methods – I propose adding missing information about the random assignment of patients to groups and the methods of measuring secondary variables.

RESPONSE: We appreciate the reviewer’s suggestion to provide additional details regarding the random assignment of patients to groups and the methods for measuring secondary variables. To address this, we have expanded the Materials and Methods section to include:

  1. Random Assignment: Details have been added regarding the randomization procedure, including the use of a 1:1 allocation ratio with block randomization (block size = 8), stratified by site to ensure balanced group assignment. The randomization sequence was maintained through sealed envelopes, ensuring blinding until statistical analysis was completed.
  2. Secondary Variables: We have elaborated on the methods used to measure secondary variables, including anthropometric parameters, plasma protein status, and contaminant monitoring. Standardized techniques for measuring anthropometry, such as head circumference, BMI, and skinfold thickness, have been described. Additionally, details about the biochemical analyses of plasma and urine biomarkers, as well as the methodologies for contaminant detection (e.g., ICP-MS for arsenic and heavy metals), have been provided. We also clarified how tolerance acquisition, digestive symptoms, and adverse events were assessed and documented.
  3. Sample Size: Additional information has been added to clarify sample size considerations, including power calculations, assumptions, and adjustments for dropout rates and expected intolerance rates. These details provide transparency on how the study population size was determined to ensure statistical robustness.

  • 3. Results – The numerical data should be presented consistently. In some parts there is a need for further interpretation of the results. The tables and figures could be explained in more detail and precision.

RESPONSE: We appreciate the reviewer’s valuable feedback, which has guided us in improving the presentation and interpretation of the results. In response, we have made the following revisions:

  1. Numerical Data Consistency:
    • Revised the baseline characteristics table to ensure uniformity in decimal precision for continuous variables and consistency in percentage formatting.
    • Adjusted all results in the text and tables to ensure alignment in units and decimal formatting.
  2. Detailed Interpretation:
    • Expanded the narrative for non-significant results, adding interpretations and clinical context to emphasize the trends observed.
    • For Z-scores and secondary anthropometric measures, we included discussions of age-related changes and their clinical relevance, even when statistical differences were not found.

  • 4. Discussion – The discussion addresses the key findings of the study, but could delve deeper into the specific findings and their broader implications. While emphasizing the statistical significance, the clinical significance of the results could be further highlighted. Although several studies are referenced, a more detailed comparison and comparison of the current results with previous results would be beneficial. The discussion could be strengthened by suggesting possible avenues for future research. The conclusions are presented cautiously but could be more concise and presented as a separate section. The conclusions could be supplemented by highlighting the practical implications of the study. We thank the reviewer for this valuable feedback, which has allowed us to refine and enhance the Discussion and Conclusions sections. In response, we have made the following revisions:
  1. Deeper Analysis of Findings:
    • We have expanded the discussion of key results, including a more detailed interpretation of the anthropometric outcomes, tolerance acquisition rates, and safety profiles of HRF and EHF. Specific clinical implications of the statistical findings, such as the comparable growth trajectories and the superior tolerance profile of HRF, have been highlighted.
    • The broader implications of these findings for CMPA management, particularly in populations with dietary restrictions, have been discussed to provide a comprehensive perspective.
  2. Comparison with Previous Studies:
    • Additional comparisons with relevant studies have been included. For instance, we discussed how our findings align with those of Fiocchi et al., Reche et al., and Nocerino et al., reinforcing the evidence for the nutritional adequacy and safety of HRF.
    • Differences and similarities between our study and prior research were outlined, emphasizing the robustness and consistency of HRF’s efficacy and tolerability across different clinical contexts.
  3. Avenues for Future Research:
    • We have suggested future research directions, including the evaluation of HRF in specific subpopulations (e.g., infants with severe CMPA or comorbidities) and long-term follow-up studies to assess growth trajectories and tolerance sustainability beyond infancy.
  4. Conclusions Section:
    • The Conclusions have been revised and presented as a separate section. We ensured the conclusions are concise and focus on the practical implications of the study, such as the suitability of HRF as a first-line dietary option for CMPA management and its potential to improve patient outcomes in real-world settings.

Round 2

Reviewer 1 Report

Comments and Suggestions for Authors

The authors tried to address the comments. However, my main comment still remains: The sample was collected from several institutions. This is not covered by the sample size calculation. You can nether identify differences between these institutions nor can you combine the samples. For that reason, I still have substantial concerns regarding the methodological rigour of this study.

Author Response

Reviewer 1:

Comments and Suggestions for Authors

The authors tried to address the comments. However, my main comment still remains: The sample was collected from several institutions. This is not covered by the sample size calculation. You can nether identify differences between these institutions nor can you combine the samples. For that reason, I still have substantial concerns regarding the methodological rigour of this study.

We appreciate your comments and your concern for the methodological rigor of our study. We would like to provide additional details regarding our sample size calculation and the procedures implemented to ensure the homogeneity of the multicenter sample. We have added this parragraph to the manuscript: Sample Size Calculation: The sample size was calculated specifically to address our primary objective. Accepting an alpha risk of 0.05 and a power of 0.8 in a two-sided test, we estimated that 53 subjects in each group would be necessary to recognize as statistically significant a difference greater than or equal to 1 unit in our primary outcome. This calculation allowed us to focus on the effect size we aimed to detect, ensuring that the sample size was adequate for our primary endpoint. Centralized Inclusion Procedure: Although our study was conducted across multiple centers, rigorous control measures were implemented to ensure the homogeneity of the sample. Compliance with inclusion and exclusion criteria was verified centrally, which minimized potential variability across institutions. Moreover, patient inclusion was non-competitive, further contributing to the homogeneity of the sample characteristics.

We hope that these clarifications address your concerns regarding our study methodology and reaffirm the robustness of our findings. We are willing to incorporate this information into the revised manuscript if deemed appropriate by the editor.

We have also added this sentence to the limitations: A larger sample would have permitted take into account covariates that may influence the development of tolerance, such as breastfeeding.

Reviewer 2 Report

Comments and Suggestions for Authors

Dear Authors, you have revised the paper very well, following the referee's suggestions.

I would like to draw attention to two points:

The first one, concerns Figure 1, as structured, with a line graph is correct, for the future probably a bar graph with scatter indices would give more value to the figure.

The second point  Regarding the sentence "Full data on all anthropometric parameters and sample sizes that allowed Z score calculations are presented by study visit in Supplementary Table 1". I mistakenly thought that the 'values' were shown in 'Supplementary Table 1', which was missing at the review stage. The table now inserted does not show the data presented with statistical significance'. I believe that it is necessary to include in the text or as supplementary material (at the authors' choice) one or more tables with the necessary information to support the numerical data (including the p-value) given in the text; this action may be useful for readers, the editor, reviewers and, above all, the authors.

Response: The authors have submitted the supplementary materials in accordance with the publication guidelines in the following section: Supplementary Materials: The following supporting information can be downloaded at …

Same consideration for “Secondary anthropometric outcome measures” Response: The authors have submitted the supplementary materials in accordance with the publication guidelines in the following section: Supplementary Materials: The following supporting information can be downloaded at …

I cannot now consult 'Supplementary Table 1' because it is not attached to the files under review (rev. 2). 

Surely the table was prepared according to the guidelines of the journal. My comment/suggestion was intended to improve the reader's consultation.

The third pointLine 354-356: The strengths of the study include its double-blind, randomised controlled design and confirmation of CMPA with double-blind OFC. Limitations include the relatively small sample size and the enrolment of infants at a relatively older age, potentially influenced by prior infant formula consumption.

Response: Thank you for your feedback. We respectfully disagree with the assertion that the sample size and the age of infant enrolment should be viewed as limitations. The sample size was calculated to ensure sufficient statistical power, considering the complex inclusion and exclusion criteria essential to maintaining methodological rigor. As for the enrolment age, the study design intentionally included infants at a range of developmental stages to capture a more comprehensive understanding of outcomes. Prior formula consumption is a realistic aspect of the target population’s background and does not detract from the study's validity. Instead, it reflects real-world conditions, making our findings more generalizable and relevant. We are confident that these aspects of the study design strengthen, rather than limit, the study’s contributions.

Perhaps also the imbalance in sample size between groups.

Response: Thank you for your observation. The allocation of participants into groups was conducted according to the calculated sample size to ensure sufficient statistical power. We carefully planned the group distribution based on sample size estimations to maintain homogeneity between groups and to optimize the study’s ability to detect meaningful differences. This approach was essential to preserve the study's internal validity and ensure that any observed effects could be attributed to the intervention rather than sample size discrepancies. We hope this addresses the concern and appreciate the opportunity to clarify our methodology.

On this point, I agree with the authors on the difficulty of forming experimental groups with a balanced number of subjects and on the fact that non-parametric statistics help researchers in statistical analysis in this type of situation; my observation was only a suggestion, as in some group comparisons the differences are substantial.

Author Response

Comments and Suggestions for Authors

Dear Authors, you have revised the paper very well, following the referee's suggestions.

Dear Reviewer thank you.

I would like to draw attention to two points:

The first one, concerns Figure 1, as structured, with a line graph is correct, for the future probably a bar graph with scatter indices would give more value to the figure.

Thank you for the suggestion, it will be considered for future submission.

The second point  Regarding the sentence "Full data on all anthropometric parameters and sample sizes that allowed Z score calculations are presented by study visit in Supplementary Table 1". I mistakenly thought that the 'values' were shown in 'Supplementary Table 1', which was missing at the review stage. The table now inserted does not show the data presented with statistical significance'. I believe that it is necessary to include in the text or as supplementary material (at the authors' choice) one or more tables with the necessary information to support the numerical data (including the p-value) given in the text; this action may be useful for readers, the editor, reviewers and, above all, the authors.

Response: The authors have submitted the supplementary materials in accordance with the publication guidelines in the following section: Supplementary Materials: The following supporting information can be downloaded at …

Same consideration for “Secondary anthropometric outcome measures” Response: The authors have submitted the supplementary materials in accordance with the publication guidelines in the following section: Supplementary Materials: The following supporting information can be downloaded at …

I cannot now consult 'Supplementary Table 1' because it is not attached to the files under review (rev. 2). 

Surely the table was prepared according to the guidelines of the journal. My comment/suggestion was intended to improve the reader's consultation.

Thank you for your comment. We have add the supplementary data to the manuscript. Sorry for the inconvenience.

The third point, Line 354-356: The strengths of the study include its double-blind, randomised controlled design and confirmation of CMPA with double-blind OFC. Limitations include the relatively small sample size and the enrolment of infants at a relatively older age, potentially influenced by prior infant formula consumption.

Response: Thank you for your feedback. We respectfully disagree with the assertion that the sample size and the age of infant enrolment should be viewed as limitations. The sample size was calculated to ensure sufficient statistical power, considering the complex inclusion and exclusion criteria essential to maintaining methodological rigor. As for the enrolment age, the study design intentionally included infants at a range of developmental stages to capture a more comprehensive understanding of outcomes. Prior formula consumption is a realistic aspect of the target population’s background and does not detract from the study's validity. Instead, it reflects real-world conditions, making our findings more generalizable and relevant. We are confident that these aspects of the study design strengthen, rather than limit, the study’s contributions.

Perhaps also the imbalance in sample size between groups.

Response: Thank you for your observation. The allocation of participants into groups was conducted according to the calculated sample size to ensure sufficient statistical power. We carefully planned the group distribution based on sample size estimations to maintain homogeneity between groups and to optimize the study’s ability to detect meaningful differences. This approach was essential to preserve the study's internal validity and ensure that any observed effects could be attributed to the intervention rather than sample size discrepancies. We hope this addresses the concern and appreciate the opportunity to clarify our methodology.

On this point, I agree with the authors on the difficulty of forming experimental groups with a balanced number of subjects and on the fact that non-parametric statistics help researchers in statistical analysis in this type of situation; my observation was only a suggestion, as in some group comparisons the differences are substantial.

Thank you for your valuable suggestion. Overall, regarding the sample size, and similarly to the Reviewer’s 1 comment, we would like to suggest to add the following sentence: “A larger sample would have permitted take into account covariates that may influence the development of tolerance, such as breastfeeding.”